

# β-Glucosidase activity and antimicrobial properties of potentially probiotic autochthonous lactic cultures

Isadora Kaline Camelo Pires de Oliveira Galdino[1,2], Miqueas Oliveira Morais da Silva[1,3], Ana Paula Albuquerque da Silva[1,3], Vanderlania Nascimento Santos[1,3], Raísa Laura Pereira Feitosa[1,3], Laura Cecília Nascimento Ferreira[1,3], Giordanni Cabral Dantas[3], Elainy Virgínia dos Santos Pereira[3], Tiago Almeida de Oliveira[4], Karina Maria Olbrich dos Santos[5], Antonio Silvio Egito[6], Flávia Carolina Alonso Buriti[1,3] and Haíssa Roberta Cardarelli[7]

[1] Centro de Ciências Biológicas e da Saúde, Universidade Estadual da Paraíba, Campina Grande, Paraíba, Brazil
[2] Centro de Tecnologia—Programa de Pós Graduação em Ciência e Tecnologia de Alimentos, Universidade Federal da Paraíba, João Pessoa, Paraíba, Brazil
[3] Núcleo de Pesquisa e Extensão em Alimentos, Universidade Estadual da Paraíba, Campina Grande, Paraíba, Brazil
[4] Departamento de Estatística, Universidade Estadual da Paraíba, Campina Grande, Paraíba, Brazil
[5] Embrapa Agroindústria de Alimentos, Empresa Brasileira de Pesquisa Agropecuária, Rio de Janeiro, Rio de Janeiro, Brazil
[6] Embrapa Caprinos e Ovinos, Núcleo Regional Nordeste, Empresa Brasileira de Pesquisa Agropecuária, Campina Grande, Paraíba, Brazil
[7] Centro de Tecnologia e Desenvolvimento Regional—Departamento de Tecnologia de Alimentos, Universidade Federal da Paraíba, João Pessoa, Paraíba, Brazil

Corresponding author
Isadora Kaline Camelo Pires de Oliveira Galdino,
isadorakaline@servidor.uepb.edu.br

## ABSTRACT

**Background:** The demand for lactic acid bacteria products, especially probiotics, has increased. Bacteria that increase polyphenol bioavailability and act as bio preservatives are sought after. This study aims to identify autochthonous lactic acid cultures from EMBRAPA that demonstrate β-glucosidase activity and inhibitory effect on microbial sanitary indicators.

**Methods:** Cell-free extracts were obtained by sonicating every 5 s for 40 min. The extracts were mixed with cellobiose and incubated at 50 °C. The reaction was stopped by immersing the tubes in boiling water. The GOD-POD reagent was added for spectrophotometer readings. Antimicrobial activity was tested against reference strains using the agar well diffusion method. Lactic cultures in MRS broth were added to 0.9 cm wells and incubated. The diameter of the inhibition zones was measured to determine the extension of inhibition.

**Results:** Only *L. rhamnosus* EM1107 displayed extracellular β-glucosidase activity, while all autochthonous strains except *L. plantarum* CNPC020 demonstrated intracellular activity for this enzyme. *L. plantarum* CNPC003 had the highest values. On the other hand, *L. plantarum* CNPC020, similarly to *L. mucosae* CNPC007, exhibited notable inhibition against sanitary indicators. These two strains significantly differed from the other five autochthonous cultures regarding *S. enterica* serovar Typhimurium ATCC 14028 inhibition ($P < 0.05$). However, they did not differ from at least one positive control in terms of inhibition against *S. aureus* ATCC

25923 and *E. coli* ATCC 25922 (*P* > 0.05). Therefore, it is advisable to consider these cultures separately for different technological purposes, such as phenolics metabolism or bio preservative activity. This will facilitate appropriate selection based on each specific property required for the intended product development.

## INTRODUCTION

Consumers are becoming increasingly conscious of the role of nutrition and health in their lives. Consequently, they actively seek healthier food options (*Barros et al., 2020*). This consumer behavior, coupled with an aging population and an increase in gut-related disorders, drives the probiotic product market. The market for probiotic products is expected to grow to approximately US$ 85.4 billion by the end of 2027 (*Market Research Report, 2022*).

Various probiotic food products are available on the market, including dairy products such as fermented milk, cheese and ice cream, and non-dairy options like cereals, fruit juice, vegetables, and meat (*Min et al., 2019*). However, using probiotic microorganisms still presents numerous challenges in terms of technological and therapeutic aspects (*Barros et al., 2020*). Moreover, the high costs associated with commercial probiotics make their use impractical for small producers and the low-income population. Therefore, it becomes crucial to study new autochthonous strains with probiotic potential (*dos Santos et al., 2015*; *Vinderola et al., 2008*).

Similar attention has been directed towards polyphenols, plant compounds that offer many benefits including antioxidant activity. However, the effects of polyphenols on human health are influenced by their metabolism within the intestinal microbiota, which plays an essential role in determining their bioavailability (*Gaya, Peirotén & Landete, 2020*).

Probiotics offer various benefits including resistance to pathogens, stabilization of intestinal microbiota after antibiotic use, increased mineral absorption, production of vitamins B and K; immune system stimulation, mutagenicity inhibition, anti-carcinogenic effects, reduced risk of colon cancer, decreased risk of cardiovascular disease, reduced serum cholesterol levels, and antihypertensive effects (*Freire et al., 2017*; *Martinez, Bedani & Saad, 2015*; *Ribeiro et al., 2014*; *Verruck, Dantas & Prudencio, 2019*).

Furthermore, the incorporation of probiotics in food can influence the taste, aroma and texture of products such as fermented milk, yogurt, cheese, fermented plant-based beverages, fruits and vegetables, fermented bread dough, and fermented meat (*de Souza, de Oliveira & de Oliveira, 2023*). Only a few studies have examined the β-glucosidase and antimicrobial activities of native lactic acid bacteria among the many probiotic studies conducted.

For example, flavonoids are glycosylated polyphenols and need to be hydrolyzed to aglycones to become bioavailable. This process is facilitated by β-glucosidases from human cells and/or microorganisms from the intestinal microbiota (*Landete et al., 2016*).

β-Glucosidases are enzymes that catalyze the hydrolysis of β-glycosidic bonds and can be produced by lactic acid bacteria, such as *Lactobacillaceae*, and bifidobacteria. Several studies demonstrate the ability of *Lactobacillaceae* strains, many of which possess probiotics properties, to produce the β-glucosidase enzyme (*Gouripur & Kaliwal, 2017*; *Ávila et al., 2009*; *Rekha & Vijayalakshmi, 2011*). This enzyme can be used to convert *O*-glycosylated phenolic compounds into bioactive aglycones, contributing to either nutritional and sensory aspects of fermented foods, such as better flavor and fragrance, or increasing bioavailability of antioxidant metabolites of plant origin (*Rokni et al., 2021*; *Michlmayr & Kneifel, 2014*; *Pérez-Martín et al., 2012*).

Probiotic strains can inhibit pathogenic and spoilage microorganisms in food matrices by producing substances like organic acids, bacteriocins, and amino acid metabolites. This helps to maintain the food's quality, flavor, and shelf life. Several studies report the efficiency of using probiotics in inhibiting pathogenic microorganisms such as *Staphylococcus* spp. and *Listeria monocytogenes* (*Buriti, Cardarelli & Saad, 2007*; *Rolim et al., 2015*; *Pisano et al., 2022*; *Kang et al., 2020*). In this way, certain probiotic cultures can be used as a natural biopreservative, thus reducing the use of chemical preservatives by the food industry (*Wu et al., 2022*; *Rolim et al., 2015*).

This study aimed to identify autochthonous lactic acid cultures from the Brazilian Agricultural Research Corporation (EMBRAPA) collection with potential for use in food. These cultures must develop β-glucosidase activity and inhibit microbial sanitary indicators *in vitro*.

## MATERIALS AND METHODS

### Autochthonous cultures of lactic acid bacteria

Five autochthonous strains of *Lactiplantibacillus plantarum* (CNPC001, CNPC002, CNPC003, CNPC004 and CNPC020), as well as *Limosilactobacillus mucosae* CNPC007 and *Lacticaseibacillus rhamnosus* EM1107, all belonging to EMBRAPA's collection, were tested.

The autochthonous strains, made available in lyophilized form, were cultured in 5 mL of De Man Rogosa and Sharpe broth (MRS; manufactured by Laboratories Conda S. A., Madrid, Spain, distributed by Kasvi, São José dos Pinhais, Brazil) at 35 ± 2 °C for 24 h for the initial activation. Soon after, a second activation was carried out by transferring 100 μL of the first activation to glass tubes containing 5 ml of MRS broth, later incubated at 35 ± 2 °C for 24 h. This second procedure was repeated frequently for the maintenance of the cultures in a way that it was always performed before each assay.

### β-glucosidase activity assay

With some modifications, the β-glucosidase activity assay was performed following the methodology outlined by *Wood & Garcia-Campayo (1990)*. Intracellular and extracellular β-glucosidase productions were determined.

Two subsequent activations of autochthonous cultures were carried out in MRS broth, prepared as a basal medium, and modified by replacing glucose with cellobiose (Êxodo Científica, São Paulo, Brazil). The cultures were centrifuged at 15 min, 3,000 rpm (3.0182 × g), (PARSEC model CT—0603; Tecnologia Laboratorial do Brasil, Santa Catarina, Brazil) and the supernatant was used to determine the extracellular β-glucosidase production. The cells were harvested in sodium citrate (50 mM, pH 4.8, Dinâmica, Espírito Santo, Brazil), then sonicated (model Ultrasonic Cleaner 2500; Unique, Indaiatuba, Brazil, 50 rpm) with intervals of 5 s for a total duration of 40 min for the enzyme release (intracellular production). A 1-mL aliquot of each cell-free extract was incubated in 1 mL of cellobiose solution for 30 min at 50 °C, stopping the reaction by immersing the tubes in boiling water for 5 min. The glucose oxidase-peroxidase reagent (GOD-POD; Biotécnica, Varginha, Brazil) was added for readings, in triplicates, in an SP-2000 spectrophotometer (Spectrum, Shanghai, China) at 500 nm. Enzymatic activity was expressed in U (µmol of product released per minute), according to Eq. (1):

$$EA = \frac{D \times Concentration\ (\mu mol/mL) \times total\ dilution\ of\ the\ reaction\ mixture\ (mL)}{time\ (minutes)} \quad (1)$$

EA = value of activity found (µmol/min or U)
Concentration (µmol/mL) of the unknown sample $(X) = Y \pm b/a$
Y = absorbance
a = angular coefficient of the line
b = linear coefficient of the line
$D$ = enzyme dilution (if necessary dilute), if the glucose concentration obtained exceeds the linearity limit of the curve
Total dilution of the reaction mixture = 2

The determination of enzyme concentration expressed in U/mL was calculated by Eq. (2):

$$[ENZ] = \frac{D \times Concentration\ (\mu mol/mL) \times total\ dilution\ of\ the\ reaction\ mixture\ (mL)}{time\ (min) \times supernatant\ vol\ (mL)} \quad (2)$$

[ENZ] = concentration enzyme (µmol/min mL or U/mL)
Concentration (µmol/mL) of the unknown sample $(X) = Y \pm b/a$
Y = absorbance found
a = angular coefficient of the line
b = linear coefficient of the line
$D$ = enzyme dilution (if necessary dilute), if the glucose concentration obtained exceeds the linearity limit of the curve
Total dilution of the reaction mixture = 2

The complete description of the β-glucosidase activity assay methodology is included in the Data S1.

## Inhibitory effect assay on reference strains of microbial sanitary indicators

The autochthonous cultures were tested for antimicrobial activity against the following standard strains of sanitary indicators: *Salmonella enterica* serovar Typhimurium ATCC 14028, *Staphylococcus aureus* ATCC 25923 and *Escherichia coli* ATCC 25922. These assays were conducted using the agar well diffusion technique (*Fernandes, de Oliveira & de Souza, 2021*).

Before carrying out the assays, the microbial sanitary indicator strains were activated from samples in brain heart infusion (BHI) broth and incubated at $35 \pm 2$ °C for 48 h. To standardize the inoculum density for the assays, the suspensions were adjusted according to the 0.5 McFarland standard, using an SP-2000 spectrophotometer (Spectrum, Shanghai, China) at 625 nm, thus obtaining a suspension containing approximately $1 \times 10^8$ CFU/mL (*Clinical and Laboratory Standards Institute, 2015*).

Subsequently, aseptically inoculated on the surface by the microorganisms with a swab previously dipped in the standardized suspension. Petri dishes with a $15 \times 2.5$ cm diameter, containing 50 mL Mueller-Hinton agar (Himedia, Mumbai, India). Then the 0.9 cm wells were filled with 50 µL of the autochthonous cultures in MRS broth. As positive controls, Ciprofloxacin 2 mg/mL (Fresoflox, Barueri, Brazil) was diluted to a concentration of 5 µg with 25 µL of the diluted solution added to the well (liquid control). A Ciprofloxacin 5 µg disk (Laborclin, Pinhais, Brazil) was also used as the control disk. The plates were incubated at $35 \pm 2$ °C for 24 h. All the procedures were repeated twice (independent duplicates). After the incubation time, for the plates that showed satisfactory traces of the inoculum and the resulting zones of inhibition (Fig. S1), the inhibition diameters were measured along with the well diameter. The calculation of the inhibition zones was used subtracting the well diameter (0.9 cm) and the values were expressed in cm.

The inhibitory effect in percentage (%) was calculated in relation to the inhibition zones of the liquid positive control and the tested strains, according to Eq. (3):

$$Inhibitory\ effect\ (\%) = \frac{HS}{HC} \times 100 \qquad (3)$$

where:

    Inhibitory effect (%) = percentage of inhibition of the tested culture
    HS = inhibition zone (cm) of tested samples
    HC = liquid positive control inhibition zone (cm)

## Statistical analysis

Results were expressed as mean ± standard deviation for the β-glucosidase assay and as mean, minimum and maximum values for the microbial inhibitory effect assay. All raw data obtained (Data S2) was submitted to one-way analysis of variance (ANOVA) to identify normality, then Fisher's least significant difference (LSD) test was performed to assess the lowest probability of significant difference between the analyzed cultures, considering $P < 0.05$ using the Statistica software version 6.0 (Statsoft Inc., Tulsa, OK, USA). Spearman's rank-order correlation between data of β-glucosidase and antimicrobial

assays were evaluated using the R software, with the Rstudio integrated development environment (IDE) for R.

## RESULTS

The results in Table 1 show that *Lactiplantibacillus plantarum* CNPC003 had the highest intracellular β-glucosidase activity (U or μmol/min) and the highest enzyme concentration (U/mL), which differed significantly from other cultures tested in the Fisher's LSD test in this present study ($P ≤ 0.034$) (Data S2). *Lactiplantibacillus plantarum* CNPC020 was the only culture that was unable to produce the intracellular enzyme, differing significantly from the others in the Fisher's LSD test ($P ≤ 1.1 × 10^{10}$) (Data S2). However, when performing the extracellular assay, *Lacticaseibacillus rhamnosus* EM1107 was the only culture that produced β-glucosidase, differing significantly from the others ($P = 0.00$) (Data S2).

The results of antimicrobial activity against *Salmonella enterica* serovar Typhimurium ATCC 14028, *Staphylococcus aureus* ATCC 25923 and *Escherichia coli* ATCC 25922 are shown in Table 2. The autochthonous lactic cultures *L. plantarum* CNPC004, *L. plantarum* CNPC020 and *L. mucosae* CNPC007 showed the highest values for *S. enterica* serovar Typhimurium inhibition. It differed significantly ($P ≤ 0.0297$) from the other *L. plantarum* strains (CNPC001, CNPC002 and CNPC003), although without significant difference from *L. rhamnosus* EM1107 ($P ≥ 0.085$) (Data S2). Although for the inhibition against *Staphylococcus aureus* ATCC 25923 there was no significant differences between the autochthonous cultures tested in the one-way ANOVA ($P = 0.107$) (Data S2), the *L. plantarum* strain CNPC002 tended to show the most significant inhibition zones with average values of 0.35 cm. However, only *L. plantarum* CNPC003 could not inhibit *E. coli* ATCC 25922. On the other hand, the *L. plantarum* strains CNPC004 and CNPC020, as well as *L. mucosae* CNPC007 and *L. rhamnosus* EM1107 were able to inhibit the three microbial indicators tested, although in different proportions.

According to the data shown in Table 2, the strains that exhibited the highest inhibition against *S. enterica* serovar Typhimurium ATCC 14028 were *L. plantarum* CNPC020 (20.62%), *L. mucosae* CNPC007 (20.87%) and CNPC004 (16.78%). However, no significant differences were observed between these strains ($P = 0.0628$) based on the one-way ANOVA (Data S2). The *L. plantarum* strains CNPC001, CNPC002 and CNPC003 had no inhibitory effect on *Salmonella enterica* serovar Typhimurium ATCC 14028. Despite not differing significantly from the other strains ($P = 0.578$) (Data S2), *L. plantarum* CNPC002 tended to show the highest percentage of inhibition against *Staphylococcus aureus* ATCC 25923, with an average of 27.97%. *L. plantarum* cultures CNPC020 and CNPC001 showed equal percentages of inhibition (50%) against *Escherichia coli* ATCC 25922, although without significant differences from the other strains ($P = 0.943$) (Data S2).

No correlations between the results for β-glucosidase and antimicrobial assays were observed for the autochthonous strains of the present study in Spearman's rank (Data S2).

**Table 1 Enzyme activity U (μmol/min) and enzyme concentration (U/mL).**

| Cultures | Intracellular | | Extracellular | |
|---|---|---|---|---|
| | Enzyme activity U (μmol/min) | Enzyme concentration (U/mL) | Enzyme activity U (μmol/min) | Enzyme concentration (U/mL) |
| *L. plantarum* CNPC001 | 0.082 ± 0.000[b] | 0.016 ± 0.001[b] | 0.000 ± 0.000[a] | 0.000 ± 0.000[a] |
| *L. plantarum* CNPC002 | 0.076 ± 0.001[b] | 0.015 ± 0.000[b] | 0.000 ± 0.000[a] | 0.000 ± 0.000[a] |
| *L. plantarum* CNPC003 | 0.094 ± 0.013[c] | 0.019 ± 0.003[c] | 0.000 ± 0.000[a] | 0.000 ± 0.000[a] |
| *L. plantarum* CNPC004 | 0.083 ± 0.000[b] | 0.017 ± 0.017[b] | 0.000 ± 0.000[a] | 0.000 ± 0.000[a] |
| *L. plantarum* CNPC020 | 0.000 ± 0.000[a] | 0.000 ± 0.000[a] | 0.000 ± 0.000[a] | 0.000 ± 0.000[a] |
| *L. mucosae* CNPC007 | 0.082 ± 0.002[b] | 0.016 ± 0.000[b] | 0.000 ± 0.000[a] | 0.000 ± 0.000[a] |
| *L. rhamnosus* EM1107 | 0.083 ± 0.001[b] | 0.017 ± 0.000[b] | 0.082 ± 0.000[b] | 0.016 ± 0.000[b] |

Note:
[a,b,c] Different lowercase letters denote that trials differ significantly ($P < 0.05$) by Fisher's LSD test (the exact $P$ values are shown in the Data S1).

**Table 2 Inhibition zones (cm) resulted by autochthonous lactic acid cultures on Mueller–Hinton agar against the reference strains of sanitary indicators (mean ± standard deviation).**

| Autochthonous lactic acid cultures | *Salmonella enterica* serovar Typhimurium ATCC 14028 | | *Staphylococcus aureus* ATCC 25923 | | *Escherichia coli* ATCC 25922 | |
|---|---|---|---|---|---|---|
| | Zone of inhibition (cm) | Inhibitory effect (%) | Zone of inhibition (cm) | Inhibitory effect (%) | Zone of inhibition (cm) | Inhibitory effect (%) |
| *L. plantarum* CNPC001 | 0.00 ± 0.00[a] | 0.00 ± 0.00[a] | 0.25 ± 0.35[b] | 20.83 ± 29.42[a] | 0.25 ± 0.07[ab] | 50.00 ± 14.14[a] |
| *L. plantarum* CNPC002 | 0.00 ± 0.00[a] | 0.00 ± 0.00[a] | 0.35 ± 0.21[b] | 27.98 ± 19.36[a] | 0.10 ± 0.14[b] | 20.00 ± 28.28[a] |
| *L. plantarum* CNPC003 | 0.00 ± 0.00[a] | 0.00 ± 0.00[a] | 0.55 ± 0.78[b] | 10.71 ± 15.15[a] | 0.00 ± 0.00[a] | 0.00 ± 0.00[a] |
| *L. plantarum* CNPC004 | 0.20 ± 0.00[a] | 16.78 ± 1.97[a] | 0.30 ± 0.00[b] | 23.21 ± 2.53[a] | 0.13 ± 0.18[b] | 25.00 ± 35.36[a] |
| *L. plantarum* CNPC020 | 0.25 ± 0.07[b] | 20.63 ± 3.46[b] | 0.33 ± 0.18[b] | 24.40 ± 10.94[a] | 0.25 ± 0.07[b] | 50.00 ± 14.14[a] |
| *L. mucosae* CNPC007 | 0.25 ± 0.06[b] | 20.87 ± 7.76[b] | 0.30 ± 0.00[b] | 23.21 ± 2.530[a] | 0.15 ± 0.21[ab] | 30.00 ± 42.43[a] |
| *L. rhamnosus* EM1107 | 0.15 ± 021[a] | 12.5 ± 17.68[a] | 0.20 ± 0.00[b] | 19.14 ± 6.86[a] | 0.11 ± 0.15[b] | 22.00 ± 31.11 |
| Positive control (Liquid) | 1.00 ± 0.00[c] | 100 | 1.30 ± 0.14[a] | 100 | 0.43 ± 0.25[b] | 100 |
| Positive control (Disc) | 1.30 ± 0.00[c] | | 0.65 ± 0.21[b] | | 0.70 ± 0.00[a] | |

Note:
[a,b,c] Different lowercase letters denote that trials differ significantly $P < 0.05$ by Fisher's LSD test (the exact $P$ values are shown in the Data S1).

## DISCUSSION

The results of the present study showed that all strains studied could produce intracellular β-glucosidase and presented enzymatic activity, apart from *L. plantarum* CNPC020. On the other hand, the one that presented the best result of intracellular β-glucosidase activity was the culture *L. plantarum* CNPC003 (0.094 U) and enzymatic concentration (0.019 U/mL). A similar result was observed by *Gouripur & Kaliwal (2017)*, that reported the production of β-glucosidase by *L. plantarum* LSP-24, but at higher concentrations (0.31 U/mL). These authors observed that the best incubation temperature to produce this enzyme was 37 °C, the same temperature used in the present study. According to these authors, several studies isolated extracellular β-glucosidase, but little attention was given to the intracellular β-glucosidase produced by *L plantarum*. Furthermore, *Rokni et al. (2021)* reported the induction of extracellular β-glucosidase production by *L. plantarum* FSO1.

However, no *L. plantarum* strain showed extracellular activity in the present study. In this study, only the *L. rhamnosus* EM 1107 culture exhibited the ability to produce extracellular β-glucosidase, which aligns with the findings of *Liu et al. (2021)*. In their study, the authors demonstrated that *L. rhamnosus* L08 had membrane-bound β-glucosidase, with enzymatic activity between 2.06 and 2.52 U/mL at different pH levels and showed the best conversion of polyphenols in apple pomace compared to the other strains also studied by these authors. *Gaya, Peirotén & Landete (2020)* reported that *Limosilactobacillus mucosae* INIA (formerly *Lactobacillus mucosae* INIA) produce β-glucosidase, which enhances the bioavailability of polyphenols through enzymatic activities. According to *Modrackova et al. (2020)*, the difference in β-glucosidase production can be attributed to strain specificity and growth conditions. Factors such as pH, temperature and carbon source can change the amount of enzyme produced (*Delgado et al., 2019*). These studies reinforce the importance of probiotic cultures that can boost the absorption of certain compounds, thus optimizing their potential health benefits. Additionally, there has been a growing interest in finding safe and natural antimicrobial substances and probiotics are a promising option due to their established health advantages. When used as bio preservatives, probiotic microorganisms can prevent the growth of harmful bacteria in food products, extending their shelf life (*Buriti, Cardarelli & Saad, 2007*; *Rolim et al., 2015*).

In this study, the strains *L. plantarum* CNPC004 and CNPC020, as well as *L. mucosae* CNPC007 and *L. rhamnosus* EM1107 were able to inhibit the three microbial indicators studied in Mueller Hinton agar. In turn, together with *L. mucosae* CNPC007, the *L. plantarum* strains CNPC020 and CNPC004 tended to show the highest percent values of inhibition against *Salmonella*, having 20.87%, 20.62%, and 16.78%, respectively. *L. plantarum* CNPC002 showed the highest percentage of inhibition against *S. aureus* ATCC 25923, with an average of 27.97%, followed by *L. plantarum* strains CNPC020 and CNPC004, *L. mucosae* CNPC007, *L. plantarum* CNPC001 and *L. mucosae* EM1107, with averages of 24.40% 23.21%, 23.21%, 20.83%, and 19.14%, respectively. Against *E. coli* ATCC 25922, both *L. plantarum* strains CNPC001 and CNPC020 achieved 50% inhibition without significant difference from the positive control ($P > 0.05$). Several studies documented in the literature have reported the bio-preservative effect of lactic acid bacteria among them. For instance, *Jabbari et al. (2017)* conducted a study where *L. plantarum*, isolated from Kouzeh cheese, demonstrated inhibitory activity against *Escherichia coli* ATCC 25922, *Staphylococcus epidermidis* ATCC 12228, *Salmonella typhi* ATCC 19430 and *Staphylococcus aureus* ATCC 25922. The inhibition zones exhibited maximum diameters of 1.13, 1.45 and 1.38 cm, respectively, also on Muller Hinton agar. *Sadeghi et al. (2022)* emphasize that the antimicrobial activity of lactic acid bacteria is a specific property of each strain.

Some other studies demonstrate the inhibitory activity of lactic acid bacteria directly in the final food product against the same sanitary indicator species of the present study. Several *L. plantarum* strains could show inhibitory activity against *S. aureus* in fish-based sausages in the study of *Speranza et al. (2017)*, similar to that was obtained *in vitro* for the *L. plantarum* strains of the present study. According to *Xu et al. (2023)*, *L. plantarum* No. 23941 can be used as a preservative in food processing instead of chemical preservatives

simultaneously against two pathogens *E. coli* and *S. aureus*. *Buriti, Cardarelli & Saad (2007)* also demonstrated in fresh cream cheese that *Lacticaseibacillus paracasei* (formerly *Lactobacillus paracasei*) LBC 82, in co-culture with *Streptococcus thermophilus* TA-40, was able to inhibit total coliforms, *Staphylococcus* spp. and *Staphylococcus* DNA positive. In another study, *Oliveira et al. (2014)* demonstrated that the probiotic microorganisms *L. acidophilus* La-5 and *L. paracasei* 01 delayed the growth of *Staphylococcus aureus* in goat cheese.

According to *Arrioja-Bretón et al. (2020)* the cell-free supernatants of the strains *L. plantarum* NRRL B-4496 and *L. rhamnosus* NRRL B-442 were able to inhibit *E. coli* (2.023 and 1.715 cm, respectively), *Salmonella* Typhimurium (2.489 and 1.889 cm respectively) and *Staphylococcus aureus* ATCC 29213 (2.053 and 2.183 cm, respectively) in tryptic soy agar. *L. plantarum* NRRL B-4496 was tested as bio preservative in beef, showing the ability to reduce *S.* Typhimurium and *Listeria monocytogenes* in this product. Moreover, several recent studies also report the antifungal activity of probiotic strains belonging to the species *L. plantarum* (*Adithi et al., 2022*; *Prabawati, Turner & Bansal, 2023*) and *L. rhamnosus* (*Chae et al., 2022*).

## CONCLUSIONS

All the autochthonous lactic acid cultures, except for *L. plantarum* CNPC020, showed intracellular β-glucosidase activity. *L. plantarum* CNPC003 had the highest activity. Bacteria expressing this activity are biotechnologically important for functional foods and bioavailable polyphenols production.

As for the inhibition of pathogens, the cultures that stood out the most were *L. plantarum* CNPC004 and CNPC020, followed by *L. mucosae* CNPC007 and *L. rhamnosus* EM1107, since all of them were able to inhibit *Salmonella enterica* serovar Typhimurium ATCC 14028, *Staphylococcus aureus* ATCC 25923 and *Escherichia coli* ATCC 25922. Beyond demonstrating inhibitory activity against the three tested pathogens, *L. rhamnosus* was the only one capable of showing extracellular β-glucosidase activity. Although *Lacticaseibacillus rhamnosus* EM1107 showed activity in all tests, none of the cultures showed maximum activity in all assays. Therefore, it is recommended to consider each culture separately for different technological purposes (metabolism of phenolics or bio-preservative activity) and direct them accordingly based on their ability to perform each specific property for the product. Future studies should use these strains as co-cultures to combine their properties and optimize food products. It is also necessary to evaluate dosage-dependent antimicrobial activity.

## ACKNOWLEDGEMENTS

The authors thank the Brazilian Agricultural Research Corporation (EMBRAPA), which provided autochthonous strains for this study, the Center of Research and Extension on Food (NUPEA/UEPB) who helped with the microbiological analyses, and the Laboratories of Phytochemistry and Basic Biochemistry (CCBS/UEPB) which helped with the β-glucosidase analysis.

### Funding

This study was suported by the Conselho Nacional de Desenvolvimento Científico e Tecnológico (CNPq, Projects N° 125627/2021-1, 147053/2022-6, 307075/2020-6 and 308253/2020-5), Coordenação de Aperfeiçoamento de Pessoal de Nível Superior (CAPES-PROAP), Fundação de Apoio à Pesquisa do Estado da Paraíba (FAPESQ, Project 028/2018) and Pró-Reitoria de Pesquisa (PROPESQ/PRPG/UFPB, internal productivity call N° 03/2020-PVM 13515-2020). This study was also financed by Paraiba State University, grant #02/2023. The funders had no role in study design, data collection and analysis, decision to publish, or preparation of the manuscript.

### Grant Disclosures

The following grant information was disclosed by the authors:
Conselho Nacional de Desenvolvimento Científico e Tecnológico: 125627/2021-1, 147053/2022-6, 307075/2020-6 and 308253/2020-5.
Coordenação de Aperfeiçoamento de Pessoal de Nível Superior (CAPES-PROAP).
Fundação de Apoio à Pesquisa do Estado da Paraíba: 028/2018.
Pró-Reitoria de Pesquisa (PROPESQ/PRPG/UFPB): 03/2020-PVM 13515-2020.
Paraíba State University: 02/2023.

### Competing Interests

The authors declare that they have no competing interests. Karina Maria Olbrich dos Santos and Antonio Silvio do Egito are employed by Brazilian Agricultural Research Corporation.

### Author Contributions

- Isadora Kaline Camelo Pires de Oliveira Galdino performed the experiments, analyzed the data, prepared figures and/or tables, authored or reviewed drafts of the article, and approved the final draft.
- Miqueas Oliveira Morais da Silva performed the experiments, analyzed the data, authored or reviewed drafts of the article, and approved the final draft.
- Ana Paula Albuquerque da Silva performed the experiments, analyzed the data, authored or reviewed drafts of the article, and approved the final draft.
- Vanderlania do Nascimento Santos performed the experiments, authored or reviewed drafts of the article, and approved the final draft.
- Raísa Laura Pereira Feitosa performed the experiments, authored or reviewed drafts of the article, and approved the final draft.
- Laura Cecília do Nascimento Ferreira performed the experiments, authored or reviewed drafts of the article, and approved the final draft.
- Giordanni Cabral Dantas analyzed the data, authored or reviewed drafts of the article, and approved the final draft.

- Elainy Virgínia dos Santos Pereira analyzed the data, authored or reviewed drafts of the article, and approved the final draft.
- Tiago Almeida de Oliveira analyzed the data, authored or reviewed drafts of the article, and approved the final draft.
- Karina Maria Olbrich dos Santos conceived and designed the experiments, authored or reviewed drafts of the article, and approved the final draft.
- Antonio Silvio do Egito conceived and designed the experiments, authored or reviewed drafts of the article, and approved the final draft.
- Flávia Carolina Alonso Buriti conceived and designed the experiments, analyzed the data, prepared figures and/or tables, authored or reviewed drafts of the article, and approved the final draft.
- Haíssa Roberta Cardarelli conceived and designed the experiments, authored or reviewed drafts of the article, and approved the final draft.

## Data Availability

The raw data are available in the Supplemental Files.

## Supplemental Information

Supplemental information for this article can be found online at http://dx.doi.org/10.7717/peerj.16094#supplemental-information.

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
