# Peer review of "β-Glucosidase activity and antimicrobial properties of potentially probiotic autochthonous lactic cultures"

_PeerJ, doi:10.7717/peerj.16094_

## Round 0.1 · original submission · Major Revisions

Please provide a comprehensively revised version addressing the editorial comments and a detailed rebuttal letter. The conclusions in the abstract are confusing; please have it thoroughly proofed for clarity.

Reviewer 1 ·

Basic reporting

This paper investigated the autochthonous lactic acid cultures from the Brazilian Agricultural Research Corporation (EMBRAPA) that have the potential to enhance food. The authors investigated five Lactiplantibacillus plantarum strains (CNPC001, CNPC002, CNPC003, CNPC004 and CNPC020), Limosilactobacillus mucosae CNPC007 and Lacticaseibacillus rhamnosus EM1107 for their β-glucosidase activity and antimicrobial effects. They found that L. rhamnosus EM1107 showed extracellular β-glucosidase activity, while all other strains, except L. plantarum CNPC020, had intracellular activity. In terms of antimicrobial effects, L. plantarum CNPC020 and L. mucosae CNPC007 showed the most significant inhibition against sanitary indicators. The study concludes that the strains should be directed towards specific technological purposes in order to obtain the best features for the final product. Overall the manuscript is well written, I only have one suggestion to make the paper better:

Introduction part, the authors discussed the importance of investigating the probiotic strains because customers are increasingly aware of the importance of diet habits. It would be great to demonstrate the importance of studying lactic acid cultures. The author briefly mentioned lactic acid bacteria can produce β-glucosidase. The introduction could be expanded to provide more context and background information on why lactic acid bacteria are important in food products and the potential benefits of using probiotics.

Experimental design

na

Validity of the findings

na

Additional comments

na

Reviewer 2 ·

Basic reporting

The manuscript could benefit from a revision of the English language.

The manuscript is well-structured. The introduction provides a relevant background on the studied topic and appropriate references are cited.

Experimental design

The aims of the study are clearly stated both in the abstract and at the end of the introduction.
The research question is well defined.
Regarding the experimental design, replicates were used to ensure the reproducibility of the experiments and adequate statistical analysis was performed.

Validity of the findings

The impact and novelty of the study need to be properly highlighted.

This concluding sentence in the abstract "Since none of the cultures obtained the best results in all tests, they should be separately directed to the desired technological purpose, whether phenolic metabolism, or antimicrobial effect, in order to obtain the best features for the final product." seems to contradict the conclusions drawn in the Conclusions section. Please clarify.

Reviewer 3 ·

Basic reporting

The manuscript provides interesting data for β-glucosidase activity and inhibitory effect on microbial indicators of five LAB strains from EMBRAPA collection. The determination of these technological characteristics is of interest for the application of the studied strains in food. The topic of the article corresponds to the field of biological sciences as Aims & Scope of PeerJ.
The described research includes an appropriate introduction and references.
The manuscript is structured according the journal format.
The language and writing style are understandable and technically correct.

Experimental design

The research goals are defined as in virto determination of significant technological parameters β-glucosidase activity and microbial inhibition, used in foods.
The method sections need significant improvement before publication. β-glucosidase activity assay should be described in more detail to be reproducible by another investigator. It is recommended to review the calculation formulas used and the units of the results obtained.
In the analysis to determine the inhibition effect, it is necessary to indicate in what units the diameter of the zones is measured and to clarify the method of calculation of the results. For example, at fig. S1 was shown, that the diameter of zone of the positive liquid control at S. aureus is not 1.3 mm, as described in Table 2.

Validity of the findings

The obtained results of the conducted analyzes are presented in three tables and are described in the results section with the applied statistical analysis.
It is recommended to review the values of the presented result regarding the units of measurement.
Also, why the data in tables are not presented as means ± standard deviation?
Do the results of Tables 2 and 3 need to be presented separately?
All numerical values in the text and in the tables should be check carefully and a decimal point should be used (for example 1.3, not 1,3).
It is necessary to carefully check the designations for fig. S1.

The discussion section can be improved by a comparative discussion of the obtained results, specifically for the inhibitory activity of the tested indicator microorganisms, with results of other authors from the last two or three years.

---

## Round 0.2 · Minor Revisions

In addition to the technical issues that have already been addressed, the manuscript could benefit from further editorial corrections. I have attached a file with some suggestions for you to review.

**Language Note:** The Academic Editor has identified that the English language must be improved. PeerJ can provide language editing services - please contact us at copyediting@peerj.com for pricing (be sure to provide your manuscript number and title). Alternatively, you should make your own arrangements to improve the language quality and provide details in your response letter. – PeerJ Staff

Reviewer 3 ·

Basic reporting

After a major revision of the article, it has been improved to a very large extent. Updated figures and tables present the obtained results more clearly.

Experimental design

No comment.

Validity of the findings

No comment.

Additional comments

It is necessary to carefully check for typographical and linguistic errors and to unify the units and designations in tables and figures (for example: on line 171, the units are described as U/ml, and in table 1, UI/ml are indicated; at figure S1 check for "Lactiplantibacillus plantarum CNPC004 (4)" and "Lactiplantibacillus plantarum CNPC020 (4)").

---

## Round 0.3 · Minor Revisions

Kindly find attached a PDF document containing some typos and suggestions that could improve your submission. I would appreciate it if you could review them and submit a revised version. Thank you for your attention to detail!

---

## Round 0.4 · accepted · Accept

Your manuscript is now accepted in PeerJ. Congratulations!